# Multimodal Imaging Based Predictors for the Development of Choroidal Neovascularization in Patients with Central Serous Chorioretinopathy

**DOI:** 10.3390/jcm12052069

**Published:** 2023-03-06

**Authors:** Sonny Caplash, Thamolwan Surakiatchanukul, Supriya Arora, Dmitrii S. Maltsev, Sumit Randhir Singh, Niroj Kumar Sahoo, Deepika Parameshwarappa, Alexei N. Kulikov, Claudio Iovino, Filippo Tatti, Ramkailash Gujar, Ramesh Venkatesh, Nikitha Gurram Reddy, Ram Snehith, Enrico Peiretti, Marco Lupidi, Jay Chhablani

**Affiliations:** 1Department of Ophthalmology, University of Pittsburgh Medical Center, 203 Lothrop Street, Pittsburgh, PA 15213, USA; 2Department of Ophthalmology, Jamaica Hospital Medical Center, 8900 Van Wyck Expy, New York Medical College, New York, NY 11418, USA; 3Princess Margaret Hospital, 3MF7+P9G, Shirley St, Nassau P.O. Box N-3730, Bahamas; 4Department of Ophthalmology, Military Medical Academy, 194044 St. Petersburg, Russia; 5Scheie Eye Institute, 51 N 39th St, Philadelphia, PA 19104, USA; 6LV Prasad Eye Institute, Kode Venkatadri Chowdary Campus, Penamaluru Rd, Tadigadapa, Vijayawada 521134, India; 7Department of Surgical Sciences, Eye Clinic, University of Cagliari, Via Università, 40, 09124 Cagliari, Italy; 8Department of Ophthalmology, University of Perugia, S. Maria della Misericordia Hospital, Piazza Università, 1, 06156 Perugia, Italy; 9Department of Retina and Vitreous, Narayana Nethralaya Foundation, 1st Main, Binnamangala, Defence Colony, 100 Feet Road, Bengaluru 560099, India

**Keywords:** central serous chorioretinopathy (CSCR), central serous, choroidal neovascularization, multimodal imaging, central serous chorioretinopathy

## Abstract

This study evaluated predictors for choroidal neovascularization (CNV) associated with central serous chorioretinopathy (CSCR) based on multimodal imaging. A retrospective multicenter chart review was conducted on 134 eyes of 132 consecutive patients with CSCR. Eyes were classified as per the multimodal imaging-based classification of CSCR at baseline into simple/complex CSCR and primary episode/recurrent/resolved CSCR. Baseline characteristics of CNV and predictors were evaluated with ANOVA. In 134 eyes with CSCR, 32.8% had CNV (*n* = 44) with 72.7% having complex CSCR (*n* = 32), 22.7% having simple (*n* = 10) and 4.5% having atypical (*n* = 2). Primary CSCR with CNV were older (58 vs. 47, *p* = 0.00003), with worse visual acuity (0.56 vs. 0.75, *p* = 0.01) and of longer duration (median 7 vs. 1, *p* = 0.0002) than those without CNV. Similarly, recurrent CSCR with CNV were older (61 vs. 52, *p* = 0.004) than those without CNV. Patients with complex CSCR were 2.72 times more likely to have CNV than patients with simple CSCR. In conclusion, CNV associated with CSCR was more likely in complex CSCR and older age of presentation. Both primary and recurrent CSCR are implicated in CNV development. Patients with complex CSCR were 2.72 times more likely to have CNV than patients with simple CSCR. Multimodal imaging-based classification of CSCR supports detailed analysis of associated CNV.

## 1. Introduction

First described by von Graefe in 1866, central serous chorioretinopathy (CSCR) is a disease characterized by a serous neurosensory retinal detachment due to choriocapillaris leakage [1,2,3,4]. The vast majority of cases of CSCR resolve spontaneously, usually within 3 to 6 months [5,6,7]. However, there is a minority of patients that progress to chronic CSCR, which is often complicated by progressive retinal pigment epithelium (RPE) atrophy, bullous retinal detachment, and choroidal neovascularization (CNV). These sequelae (particularly CNV) offer a poor visual prognosis for patients. Conventionally, the incidence of CNV was estimated to be from 2–18% [7,8]. Investigations examining chronic CSCR patients found that roughly one-third will go on to develop CNV [9,10]. The advent of optical coherence tomography angiography (OCTA) has demonstrated an incidence closer to 30% [11]. This is likely due to the increased sensitivity and specificity of OCTA relative to fluorescein angiography and its overall convenience as a non-invasive imaging modality [11,12].

OCTA alone, however, may not be sufficient in detecting all cases of CNV in patients with CSCR [13]. Multimodal imaging often serves as the most comprehensive approach in detecting and characterizing neovascularization in patients with CSCR. The pathophysiologic mechanism for the development of CNV in patients with CSCR is postulated to involve decompensation of the RPE with subsequent disruption of Bruch’s membrane; reminiscent of the proposed mechanism in age-related macular degeneration (ARMD) [4,11,14,15,16]. Previously, CNV associated with CSCR has been characterized broadly as a downstream sequela of the disease, with the assumption that all CNV associated with CSCR are similar in their presentation, prognosis, and effective treatment. Some studies have investigated long-term outcomes in patients with CSCR, finding that the majority of patients develop a Type I CNVM, with roughly 69 to 80% requiring either anti-VEGF or PDT [17,18,19].

Our group has recently published a revised classification that functions to standardize the terminology surrounding CSCR, allowing for more nuanced explorations into the diagnosis, management, and prognosis of the disease [16]. Our classification begins with classifying cases as simple, complex, and atypical based principally on RPE morphology. Simple and complex are differentiated by a size threshold of 2-disc diameters of clinically detectable retinal pigment epitheliopathy. Atypical cases served to include those cases of CSCR in which other retinal pathologies were also present as well as bullous CSCR. In order to further standardize the time course of CSCR, simple and complex cases were further reclassified, based on clinical course, as primary, recurrent, or resolved, and subsequently qualified as a persistent subtype [20].

Given the poor visual prognosis of patients with CSCR complicated by neovascularization, it is of particular clinical interest to stratify the risk of CNV development in patients with CSCR [4,5,7,14]. The purpose of this investigation is to establish the incidence of CNV development in relation to our recently described classification system and treatment outcome at one-year follow up for CNV associated with CSCR based on multimodal imaging

## 2. Materials and Methods

This was a retrospective, multicenter study on patients with a known diagnosis of CSCR. The study adhered to the tenets of the Declaration of Helsinki and ethical clearance was obtained by the Institutional Review Board. Informed consent was obtained from all patients.

Retrospective data were gathered from the Departments of Ophthalmology across multiple centers in the U.S., Italy, India, and Russia from 2015–2020. The charts of all patients with a diagnosis of CSCR were evaluated. Subsequent inclusion criteria were imposed, including (i) availability of demographic, clinical and reliable treatment details; (ii) availability of good quality multimodal imaging including fundus autofluorescence (FAF), spectral domain optical coherence tomography (SD OCT) (B scan), and OCTA or fundus fluorescein angiography (FFA) with indocyanine angiography (ICGA). Importantly, image quality on OCT and OCTA must have exceeded a signal strength of 7/10. With regard to angiographic imaging, patients must have had a full sequence of early, middle, and late phases without motion or lid artifacts. Exclusion criteria included were (i) any other retinal disease such as ARMD or any other cause of macular neovascularization, and (ii) poor quality imaging as discussed earlier.

Baseline data collected from patients included age, sex, best corrected visual acuity (BCVA), duration of complaints, reliable history of any previous such episodes, previous retinal treatment, or steroid use.

FAF, fundus photographs, FFA, and ICGA were obtained from Spectralis HRA + OCT (Heidelberg Engineering, Heidelberg, Germany) or F-10 scanning laser ophthalmoscope (NIDEK, Gamagori, Japan). OCTA examinations were performed with the RTVue-XR Avanti (Optovue, Fremont, CA, USA) or Spectralis HRA + OCT. For each eye, horizontal raster pattern scan through the center of the macula was obtained. OCTA examination including a 6 × 6 mm (2 orthogonal volumes with 400 × 400 A scans) pattern centered in the center of the fovea was performed with RTVue-XR Avanti.

Double blind classification was performed by two retinal experts [SA and DM] as per the new multimodal imaging-based classification system of CSCR. In cases of non-consensus, senior investigator (JC) was consulted. All images were made available by SC for all graders with images deidentified and randomly assigned by numeric generator to either expert. Eyes were classified as per the multimodal imaging-based classification of CSCR at baseline into (i) simple/complex CSCR, (ii) primary episode/recurrent/resolved CSCR, (iii) persistent SRF (>6 months) or not, (iv) outer retinal atrophy (ORA) presence or absence, (v) foveal involvement presence or absence, (vi) CNV presence or absence. Representative cases of simple and complex CSCR are illustrated in Figure 1 and Figure 2, respectively.

All patients selected by the above criteria were classified as previously published by our group [20]. Parameters evaluated at baseline, 3 months, 6 months, and 12 months included BCVA, subfoveal choroidal thickness (SFCT), and central macular thickness (CMT). Presence of CNV and treatment instituted for each of the above time intervals was also noted. Statistical analysis, including baseline characteristics of CNV and predictors, were evaluated with one-way analysis of variance (ANOVA), chi-square test, and odds ratio calculation via Microsoft Excel.

## 3. Results

Included in this analysis were 134 eyes from 132 patients with six eyes having non-consensus requiring consultation of senior investigator JC. The median age of patients was 48 years (range: 23–67). There was a strong male gender predisposition with 84.6% being male. Seven eyes had previous histories of treatment at presentation: five eyes with focal laser, one eye with photodynamic therapy (PDT), and one patient on Eplerenone.

Of the 134 eyes with CSCR, CNV was present in 32.8% (*n* = 44), and demographic data for this group is summarized in Table 1. Of all patients with CNV, 15% were detected primarily by OCT-A, though ICG-A and FA showed suspicious lesions. Of the 44 patients with CNV, 32 had complex CSCR (72.7%), 10 had simple CSCR (22.7%), and 2 had atypical CSCR (4.5%). Demographic breakdown and visual acuity per each CSCR subtype in patients with CNV is summarized in Table 2. Among the 90 patients without CNV, 50 had simple CSCR (55.6%) and 40 had complex CSCR (44.4%). There was no statistically significant difference in the prevalence of simple CSCR between patients with CNV and those without (*p* = 0.08). Patients with complex CSCR were 2.72 times more likely to have CNV than patients with simple CSCR (95% CI 1.199 to 6.166; *p* = 0.016). Table 3 summarizes the incidence of CNV among CSCR subtypes.

Patients with simple CSCR with CNV had better baseline BCVA than those with complex CSCR (0.72 vs. 0.49, *p* = 0.03) without differences in age (mean 59 vs. 59 years, *p* = 0.76), SFCT 393 vs. 390 μm, (*p* = 0.997), and reported duration of symptoms (median 6 vs. 18 weeks, *p* = 0.38).

Based on presentation, there was an equal incidence of CNV in both primary and recurrent cases (*n* = 20, 45.5% each) with the remainder of CNV found in resolved cases (9.1%, *n* = 4). Data is summarized in Table 4. There was no age-related, statistically significant difference between primary, recurrent, and resolved cohorts, with the mean age being 58 ± 10.8, 61 ± 9.8, and 62 ± 10.8, respectively (*p* = 0.27). BCVA was also not statistically significantly different among primary, recurrent, and resolved cases of CSCR with mean BCVA Snellen equivalent(logMAR) of 20/73 (0.56, standard deviation [SD] 0.29), 20/63 (0.50, SD 0.31) and 20/89 (0.65, SD 0.33), respectively (*p* = 0.66). Demographic and visual acuity data are summarized in Table 3. SFCT did not differ among these cohort as well with their mean values as 414 ± SD, 383 ± SD, and 329 ± SD μ, respectively (*p* = 0.55). Differences in overall duration among the three cohorts were not statistically significant with median values of 7 and 30 months, respectively (*p* = 0.42). A summary of duration in months across various sub-groups is listed in Table 5.

In comparing patients with primary CSCR with CNV to those without CNV, the mean age was 58 years (SD 10 years) as compared to 47 years (SD 10 years) with a *p*-value of 0.00003. The mean BCVA logMAR (Snellen fraction) was 0.75 (20/112) as compared to 0.56 (20/73) with a *p*-value of 0.01. The average duration was 7 months as compared to 1 month (*p* = 0.0002).

When comparing patients with recurrent CSCR with CNV to those without, there was no statistically significant difference in mean logMAR BCVA (0.50 vs. 0.61, *p* = 0.31), duration (median 7 vs. 2 months, *p* = 0.32), and mean SFCT (383 ± 75 vs. 424 ± 85 μm, *p* = 0.21). An exception to this trend of similarity between the two groups is in the mean age of presentation, which was 60.7-years-old in patients with recurrent CSCR with CNV and 52-years-old in recurrent CSCR patients without CNV (*p* = 0.003).

Forty-three out of the forty-four patients with CNV had foveal involvement, compared to the eighty-five out of ninety patients without CNV, which was not statistically significant (*p* = 0.38). In comparing foveal involvement between patients with CNV and patients without CNV, there was no statistically significant difference. Regarding ORA, among all CSCR patients with CNV, 36 (78%) had ORA as opposed to 29 (32%) among CSCR patients without CNV (*p* < 0.01).

## 4. Discussion

Our investigation yields several key risk factors for the development of CNV in patients with CSCR. Patients with complex CSCR were at a significantly higher risk for CNV with an odds ratio of 2.72. In both primary and recurrent CSCR, there was a significant difference in age of presentation when comparing patients that developed CNV with those who did not. Specifically, in both primary and complex CSCR, older patients were more likely to develop CNV. Our data also intuitively demonstrate that primary CSCR patients with CNV are predisposed to longer courses when compared to patients without CNV. The relative higher incidence of CNV in patients with primary CSCR compared to prior published data implies that a lower threshold should exist for use of OCTA or FA to evaluate for CNV in CSCR patients with SRF on initial presentation. Counterintuitively, our data did not support statistically significant differences in duration of disease course between primary, recurrent and atypical CSCR. This finding supports the idea that recurrence functions as a secondary measure more so than a primary prognostic measure.

The positive correlation between age upon presentation and the incidence of CNV can serve as an important determinant of pre-test probability for the development of CNV when evaluating patients with CSCR on initial work-up. These findings are in line with other studies of CSCR which found older patients with higher incidence of CNV and worse visual acuity [7,21].

Multimodal imaging-based classification provides objective approach to associate disease severity with CNV formation and progression. More broadly, increased CNV in patients with complex CSCR when taken in conjunction with our findings of increased ORA in patients with CNV, reinforce the notion of complex CSCR as retinal pigment epitheliopathy. The co-occurrence of ORA with CNV has been validated by other groups as well [22]. The importance of the association between CNV and complex CSCR has several important implications beyond elucidation of underlying pathophysiology. In the affected eye, CNV has been shown previously to be associated with cystoid macular degeneration, a marker of progressive CSCR damage [15]. Given that CSCR has been frequently demonstrated to progress to bilateral disease, it is reasonable that in patients with unilateral neovascular CSCR, there is a risk of CNV secondary to CSCR in the fellow eye. Previously published work by our group has shown that the fellow eye in patients with unilateral CSCR complicated by CNV can have early signs of neovascularization as detected by OCTA [14].

CSCR often takes an uncomplicated, self-resolving course, however, in patients in which this does not occur, morbidity and visual prognosis can be poor. Our data point towards a model of CSCR management in which early classification can allow for appropriate risk stratification with regard to the development of CNV and consequently appropriate screening via multi-modal imaging. The rise of OCTA, taken together with ICGA, FFA, and OCT allow for a layered approach to the early diagnosis of CNV in both the affected eye and the fellow eye, with early treatment mitigating complications of long-standing CSCR such as cystoid macular degeneration.

While our study provides nuance to the progression and prognosis of patients with CNV, it is limited by a relatively small sample size and retrospective nature. Studies of CSCR are often limited by the relatively low incidence. Our study’s retrospective nature precludes an ability to assess patients at a long-term follow-up. This can create bias in the data towards patients that may develop CNV later in their course. Additionally, while efforts were made to ensure randomization of patient images for classification, the retrospective nature of this study provides an inherent limitation to the completeness of randomization. It is also important to note that CSCR with CNV is often confounded by the presence of pachychoroid neovasculopathy; this element may confound interpretation of patient presentation.

The investigation and subsequent data presented tie our previously published classification system to objective, discrete outcomes in patients with CSCR, specifically, the development of CNV. Age upon presentation and complex CSCR phenotype serve as early prognostic indicators to the development of CNV and consequently poorer visual outcomes. The importance of these findings is their purported role in driving more targeted treatment for patients with CSCR. Further studies will be conducted to prospectively evaluate patients’ treatment response and to further explore the relative distribution of different types of CNV across different CSCR classifications.

## Figures and Tables

**Figure 1 jcm-12-02069-f001:**
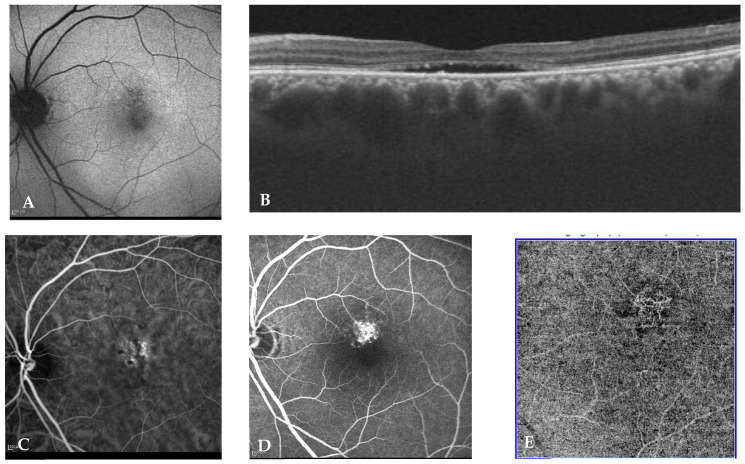
Eye of a 52-year-old male with a visual acuity of 20/20 and metamorphopsia for 4 months (**A**) shows <1–2 DD of RPE abnormality (simple CSCR) (**B**) OCT (B scan) passing through the fovea shows subretinal fluid along with flat irregular pigment epithelium detachment suspicious of CNV (**C**), (**D**,**E**) show ICG-A, FA and OCTA, respectively, all of which confirm the choroidal neovascular complex.

**Figure 2 jcm-12-02069-f002:**
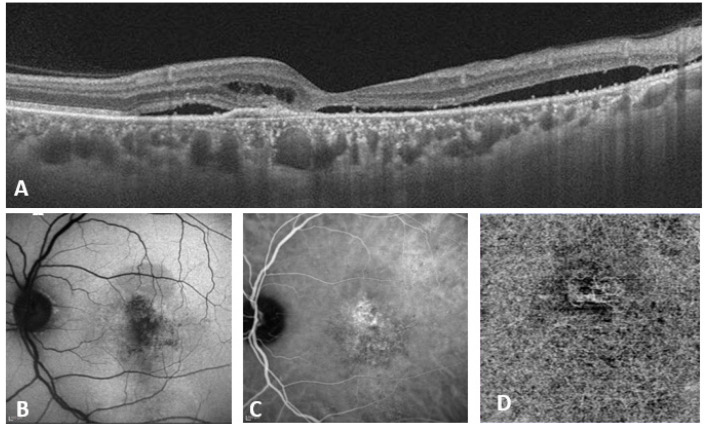
Eye of a 47-year-old male with a visual acuity of 20/320 and duration of complaints for 2 years. (**A**) OCT (B scan) passing through the fovea shows intraretinal and subretinal fluid along with flat irregular pigment epithelium detachment, suspicious for likely CNV. (**B**) Demonstrates > 2 DD area of RPE abnormality (Complex CSCR). (**C**,**D**) show corresponding ICG-A and OCTA, respectively, that confirm the choroidal neovascular complex in the outer retina and choriocapillaris slab.

**Table 1 jcm-12-02069-t001:** Choroidal Neovascularization in Central Serous Chorioretinopathy: Initial Presentation (N = 44).

Feature	Cases (%)
Age, years	59
Mean (median, range, 95% CI)	(58, 40–82, 37–81)
Age distribution	
<45	4 (9)
45–60	22 (50)
60–75	16 (36)
>75	2 (5)
Gender	
Male	37 (84)
Female	7 (16)
Visual acuity (decimal)	0.54
Mean (median, range)	(0.50, 0.06–1.20)

**Table 2 jcm-12-02069-t002:** Demographics of CNV and Non-CNV in Central Serous Chorioretinopathy: Multimodal Imaging-Based Classification: Simple, Complex vs. Atypical.

	Simple		Complex		Atypical	
	Non CNV (N = 50)	CNV (N = 10)	Non CNV (N = 40)	CNV (N = 32)	Non CNV (N = 0)	CNV (N = 2)
Mean Age, years (N)	47	59	49	59		53
Median (range)	47 (23–67)	56 (45–77)	49 (32–67)	59 (40–82)		53 (48–58)
Age distribution (%)						
<45	21 (42)	0	12 (30)	4 (13)		
45–60	22 (44)	6 (60)	20 (50)	14 (44)		2 (100)
60–75	7 (14)	3 (30)	8 (20)	13 (41)		
>75	0	1 (10)	0	1 (2)		
Gender (%)						
Male	35 (70)	8 (80)	38	27 (84)		2 (100)
Female	15 (30)	2 (20)	2	5 (16)		
Mean Visual acuity (decimal)	0.73	0.72	0.67	0.49		0.5
Median (range)	0.80 (0.03–1.20)	0.67 (0.30–1.20)	0.70 (0.06–1.20)	0.50 (0.06–1.00)		0.50 (0.33–0.66)

**Table 3 jcm-12-02069-t003:** Incidence of CNV among CSCR sub-types (% of total patients).

	Primary	Recurrent	Resolved	Total
Simple	6 (12%)	4 (50%)	0	10 (17%)
Complex	13 (36%)	15 (48%)	4 (100%)	32 (43%)
Atypical	1 (100%)	1 (100%)	0	2 (100%)
Total	20 (22%)	20 (50%)	4 (100%)	44 (33%)

**Table 4 jcm-12-02069-t004:** Demographics of CNV and Non-CNV in Central Serous Chorioretinopathy: Onset Status Multimodal Imaging-Based Classification: Primary and Recurrent vs. Resolved.

	Primary		Recurrent		Resolved	
	Non CNV (N = 69)	CNV (N = 20)	Non CNV (N = 20)	CNV (N = 20)	Non CNV (N = 1)	CNV (N = 4)
Mean Age, years	47	58	52	61	69	52
Median (range)	47 (23–67)	57 (44–77)	50 (39–65)	59 (42–82)	69	50 (40–67)
Age distribution (%)						
<45	33 (48)	1 (5)	2 (10)	1 (5)		2 (50)
45–60	29 (42)	11 (55)	12 (60)	10 (50)		1 (25)
60–75	8 (11)	7 (35)	6 (30)	8 (40)	1 (100)	1 (25)
>75	0	1 (5)	0	1 (5)		0 (0)
Gender (%)						
Male	55 (80)	15 (75)	20 (100)	17 (85)	1 (100)	4 (100)
Female	14 (20)	5 (25)		3 (15)		0 (0)
Mean Visual acuity (decimal)	0.75	0.56	0.61	0.5	0.0625	0.65
Median (range)	0.80 (0.03–1.20)	0.60 (0.10–1.00)	0.56 (0.10–1.20)	0.50 (0.06–1.20)	0.0625	0.68 (0.25–1.00)

**Table 5 jcm-12-02069-t005:** Median Duration in Months of Central Serous Chorioretinopathy per Classification and presence of CNV (N).

		Primary	Recurrent	Resolved
Simple	Non-CNV	1 (31)	1.5 (4)	
	CNV	7 (5)	4 (3)	
Complex	Non-CNV	4 (15)	60 (4)	
	CNV	12 (7)	7 (4)	30 (2)
Atypical	Non-CNV			144 (1)
	CNV	2 (1)	7 (1)	

## Data Availability

Not applicable.

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
