# Peer review of "Multimodal Imaging Based Predictors for the Development of Choroidal Neovascularization in Patients with Central Serous Chorioretinopathy"

_jcm, 2023, doi:10.3390/jcm12052069_

Round 1

Reviewer 1 Report

The manuscript is well-written and contains valuable findings regarding the trending use of OCTA in detecting CNV in patients with CSR according to the classification of a condition with a wide spectrum of phenotypes. The statistics appear sound and various findings confirm other studies. Few points that may be worth reviewing or expanding to help clarify certain points for the readers are listed below.

Introduction

“Conventionally, the incidence of CNV was esti mated to be from 2-9%7 but advent of Optical coherence tomography angiography (OCTA) has demonstrated an incidence closer to 30%8.”

·       The referenced study included (small sample of) patients with chronic CSR (not all types of CSR)- please ensure this is mentioned as can be misleading/misinterpreted

o   Sulzbacher et al also had similar findings in patients with chronic CSR

o   Savastano et al found much higher incidence in chronic CSR than “acute” CSR

·       Is this because OCTA has higher sensitivity compared to FA?  Or because OCTA is quick, convenient, non-invasive thus more widely used (Bansal et al)

·       Please mention the clinical significance of these CNV identified- do all require treatment, or a certain proportion?  With PDT/anti-VEGF

“CSCR is postulated to involve decompensation of the RPE with subsequent disruption of Bruch's membrane; reminiscent of the proposed mechanism in age-related macular degeneration

(ARMD)4, 8-“

·       When discussion RPE decompensation and subsequent disruption of Bruch’s as a mechanism for development of CNV?, was this proposed for chronic CSR?

·       Can you tie this pathophysiology in with your findings in the discussion?

o   It may be confusing for readers when discussing choriocapillaris ischaemia (or blood flow dysregulation) later in the discussion and separately mentioning RPE decompensation as the mechanism in the introduction.

“Previously, CNV associated with CSCR has been characterized broadly as a downstream sequela of the disease, with the assumption that all CNV associated with CSCR are similar in their presentation, prognosis and effective treatment”

·       Please mention any studies that distinguish type I vs type II CNV in these patients

Methods

Were other OCT factors assessed? May be worth mentioning OCT biomarkers of CNV found in other studies in the discussion. Would these also prompt the clinician to perform OCTA (other than just age, duration of primary, complex etc)

·       Subretinal fibrin

·       PED- size/, flat irregular PEDs (FIPED), double-layer sign

·       Dilated choroidal vessels

·       ORA was mentioned and consolidates findings to other studies (ONL thinning, ELM/EZ disruption, RPE attenuation)

Is there any information on inter-grader agreements among the 2 masked graders.

Although the classification has been previously validated with fair-moderate intra-grader agreement, the double-blind classification method was not detailed in this study (how many images needed were of non-consensus?). A mention in the limitations section of this system could be added (but not overly necessary).

Results

It would be nice if a table demonstrating the proportion of patients that were found to have CNV according to the CSR classification. Perhaps similar to the table in Fig 1 of editorial by Chhablani et al, “Multimodal Imaging-Based Central Serous Chorioretinopathy Classification”. The readers may more easily understand the proportion of patients that developed CNV within their classification.

Discussion

Many patients with CSR and CNV are thought to have pachychoroid neovasculopathy. Please mention that these patients were excluded or mention in the limitations that there may be possible disease overlap in the patient sample, particularly with the findings of CNV in patients with primary CSR

“precludes an ability to assess patients at long-term follow-up”

·       This could perhaps be a source of potential bias for those that may develop CNV (duration/recurrence) and should be mentioned.

“Age on presentation and complex CSCR phenotype serve as early prognostic indicators to the development of CNV and consequently poorer visual outcomes. The importance of these findings is their purported role in driving more targeted treatment for patients with CSCR.”

·       An important finding of the study was that patients with primary CSR had a relatively high incidence of CNV when conventionally it was believed to be a sequelae of chronic disease. May be worth reinforcing their findings that OCTA may be useful in those presenting with primary CSR/first known episode of SRF for detecting CNV

Author Response

Thank you kindly for your thoughtful comments and analysis, attached is our response. 

Reviewer 2 Report

This is a well written paper, however, several articles have already shared the similar results. The authors should cite PMID: 34116008 (Ng DS, et al. Am J Ophthalmol. 2021) or others. Overall, the results shown in this paper were not new.

In addition, my major concern is the study design. In the introduction, the authors wrote “Given the poor visual prognosis of patients with CSCR complicated by neovascularization, it is of particular clinical interest to determine predictive risk factors for its development”, however, the authors compared the clinical findings of eyes with CNV to the eyes without CNV, therefore, the results just showed the factors of suggesting the concomitant CNV in the eyes with CSCR, not true risk factors of the development of CNV. For this purpose, the papers such as PMID: 35537029 and PMID: 33342759 should be referred.

Author Response

Thank you for your thoughtful comments, attached is our response. 

Reviewer 3 Report

I have reviewed with interest the manuscript entitled: “Multimodal imaging based predictors for the development of choroidal neovascularization in patients with Central Serous Chorioretinopathy”.

The Authors assessed the predictors for choroidal neovascularization (CNV) associated with central serous chorioretinopathy (CSCR) by multimodal imaging in a retrospective multicenter study. They found prognostic information which may be useful to better manage patients affected by CSCR in the long term. 

The study is original and the topic is of great clinical relevance. However, some points should be carefully addressed.

Namely:

- Methods section needs to be reviewed and implemented:

1) Inclusion criteria should be more detailed

2) The authors did not state the clinical departments involved in the study

3) The authors did not state the timing of patients enrollment in this study

4) The description of the statistical analysis should be implemented

- The number of references may be improved

- Because of the clnical relevance of outer retinal atrophy (ORA) the authors may provide an additional image in order to better show ORA by OCT and FAF

Author Response

Thank you for your thoughtful review and comments, attached is our response. 

Round 2

Reviewer 2 Report

The authors adequately revised the introduction. However, the results other than the classification system are still lacking novel findings. The authors should strengthen the merit using their classification system to predict CNV. Considering the high rate of CNV in the complex CSC group, their criteria of complex CSC (total area of RPE alteration > 2 DA or multifocal) may imply the coexistence of CNV in eyes with CSC history. Could the authors elucidate which item is more important, large RPE alteration or multifocal (or both) to predict CNV?

Additional comments are as follows.

1.    Methods section. I think the presence of CNV should be confirmed using OCTA because it is well known that FA alone cannot differentiate CNV effectively from the diffuse RPE atrophy as seen in the complex CSC cases. It would be helpful for readers to know how many CNVs were diagnosed with OCTA.

2. According to their previous report, foveal involvement includes SRF, ORA and PED. However, these components originate from different conditions and when flat PED is observed at the fovea, it strongly suggests the MNV. Please indicate the details of foveal involvement.

3.    In the methods section, they noted that parameters evaluated at baseline, 3 months, 6 months and 12 months included BCVA, subfoveal choroidal thickness (SFCT), Central macular thickness (CMT), but those results were not shown.

4.    I recommend the authors revise table 1. Please compare the characteristics including the CSC classification between eyes with and without CNV.

5.    In the table 2, please indicate the total number of eyes in each category of the classification system to know the percentage of eyes with CNV in each group. If the authors would like to highlight their classification, table 4 should be incorporated into table 2.

6.    Table 3. Even eyes with primary episode might have persistent SRF. I would like to know whether the chronicity is important to predict the CNV or not.

7.    Figure 1. Please add the caption of A-C within the figure. In both figures, please add the FA and ICGA images. I suspect the polypoidal lesions in a case of figure 1. The legends should be more specified. Is the top left image fundus autofluorescence?  

Author Response

We thank Reviewer 2 for their thoughtful comments. We have addressed them and our response can be found in the attached document. 
